# *Veronica austriaca* L. Extract and Arbutin Expand Mature Double TNF-α/IFN-γ Neutrophils in Murine Bone Marrow Pool

**DOI:** 10.3390/molecules25153410

**Published:** 2020-07-28

**Authors:** Petya A. Dimitrova, Kalina Alipieva, Tsvetinka Grozdanova, Milena Leseva, Dessislava Gerginova, Svetlana Simova, Andrey S. Marchev, Vassya Bankova, Milen I. Georgiev, Milena P. Popova

**Affiliations:** 1Department of Immunology, The Stefan Angeloff Institute of Microbiology, Bulgarian Academy of Sciences, bl. 26 Acad. Georgi Bonchev Str., 1113 Sofia, Bulgaria; petya_dimitrova@web.de (P.A.D.); mlesseva@gmail.com (M.L.); 2Institute of Organic Chemistry with Centre of Phytochemistry, Bulgarian Academy of Sciences, bl. 9 Acad. Georgi Bonchev Str., 1113 Sofia, Bulgaria; tgrozdanova@orgchm.bas.bg (T.G.); dpg@orgchm.bas.bg (D.G.); sds@orgchm.bas.bg (S.S.); bankova@orgchm.bas.bg (V.B.); popova@orgchm.bas.bg (M.P.P.); 3Laboratory of Metabolomics, The Stephan Angeloff Institute of Microbiology, Bulgarian Academy of Sciences, 139 Ruski Blvd., 4000 Plovdiv, Bulgaria; andrey.marchev@yahoo.com (A.S.M.); milengeorgiev@gbg.bg (M.I.G.)

**Keywords:** *Veronica austriaca*, arbutin, neutrophils, TNF-α, IFN-γ, cell maturation

## Abstract

Plants from the *Veronica* genus are used across the world as traditional remedies. In the present study, extracts from the aerial part of the scarcely investigated *Veronica austriaca* L., collected from two habitats in Bulgaria—the Balkan Mountains (Vau-1) and the Rhodopi Mountains (Vau-2), were analyzed by nuclear magnetic resonance (NMR) spectroscopy. The secondary metabolite, arbutin, was identified as a major constituent in both extracts, and further quantified by high-performance liquid chromatography (HPLC), while catalpol, aucubin and verbascoside were detected at lower amounts. The effect of the extracts and of pure arbutin on the survival of neutrophils isolated from murine bone marrow (BM) were determined by colorimetric assay. The production of cytokines—tumor necrosis factor (TNF)-α and interferon (IFN)-γ was evaluated by flowcytometry. While Vau-1 inhibited neutrophil vitality in a dose-dependent manner, arbutin stimulated the survival of neutrophils at lower concentrations, and inhibited cell density at higher concentrations. The Vau-1 increased the level of intracellular TNF-α, while Vau-2 and arbutin failed to do so, and expanded the frequency of mature double TNF-α^+^/IFN-γ^hi^ neutrophils within the BM pool.

## 1. Introduction

Genus *Veronica* L. (speedwell) belongs to the Plantaginaceae family with ecologically diverse distribution of its plant species and disputable member affiliation, depending on synonym name recognition. The accepted species included in The Plant List are 234. In Bulgarian flora, 38 species have been described so far [1], among which, *Veronica officinalis* (common speedwell) is the most popular medicinal plant. It is used for the treatment of bruises, rheumatism, pain, cold, fracture, irregular menstruation, traumatic bleeding, embolism, catarrh, wound healing and hypercholesterolemia in Balkan traditional medicine [2], and it is included in the commercial herbal products from several European countries [3]. Another popular medicinal species is *Veronica austriaca* L. (Austrian speedwell, synonym *V. jacquinii*), used for preparing tea against cough and for the regulation of digestion in the region of the Western Balkan Peninsula [4].

Initial phytochemical studies of the genus have been performed from a chemotaxonomic point of view, reporting the presence of iridoid glycosides, phenylethanoid glycosides and acylated flavonoids as chemotaxonomic markers [5,6,7,8,9]. Although the chemistry of *Veronica* has been investigated extensively, and over 120 individual compounds have been isolated, extracts of *V. austriaca* have previously been analyzed only by thin layer chromatography-densitometry, and basic iridoid content has been reported [10]. In addition, an aqueous acetone extract from aerial parts of *V. jaquinii* has been studied by high-perfomance liquid chromatography with tandem mass spectrometry and various-compounds were found, among which chlorogenic and quinic acids, hyperoside and isoquercetin have been reported as dominant constituents [11]. Recently, Zivkovic et al. 2017 [12] analyzed methanolic extract from the same species by applying a similar technique and the main detected flavonoids appeared luteolin, isoscutelarein and quercetin derivatives, and verbascoside (acteoside), forsythoside B, forsythenside K and leucoseptoside A were reported as major phenylethanoid glycosides.

As part of our ongoing project on the metabolite profiling of *Veronica* plant species from Bulgaria and eventual biological activity of the extracts and their secondary metabolites [13,14], we undertook investigations on *V. austriaca.* Two samples of the species collected from different habitats were studied by application of the proton (^1^H) NMR in combination with two dimensional (2D) NMR techniques. We have identified arbutin as a major constituent in both extracts and evaluated the biological activity of the extracts and arbutin on the vitality of murine neutrophils derived from bone marrow, on their maturation state and ability to produce key cytokines, such as TNF-α and IFN-γ.

## 2. Results

### 2.1. Identification of Metabolites in Veronica austriaca Extracts

In the present report, we describe the analysis of *V. austriaca* collected from two different habitats, with 4 biological replicates each. We applied ^1^H NMR in combination with 2D NMR techniques (*J*-resolved, COSY, HSQC) of the methanolic extracts. In total, 13 individual compounds were assigned to the abundant signals, including amino acids, organic acids, iridoid glucosides and phenolic compounds (Table 1).

The major compound in extracts of both samples was identified as arbutin, a hydroquinone derivative-4-hydroxyphenyl-β-glucopyranoside. The most intensive doublets in ^1^H NMR spectra at δ 7.00 and δ 6.79 ppm (d, 8.9 Hz), corresponding to AA’BB’ system and a single doublet of anomeric proton of β-glucose moiety at δ 4.85 ppm (d, 7.9 Hz), were assigned to this phenolic glucoside and were confirmed by comparison with an authentic sample (Figure 1, Table 1). The presence of catalpol, aucubin and verbascoside in extracts was confirmed accordingly. Furthermore, the presence of arbutin in both extracts was determined quantitatively by HPLC, and the content of 308.19 ± 20.67 µg/mg and 382.93 ± 28.11 µg/mg extract for Vau-1 and Vau-2 was established, respectively. Table 2 presents the content of arbutin in various concentrations of the extracts applied in our study.

### 2.2. Effect of the Vau-1/Vau-2 Extracts or Arbutin on Vitality of BM-Derived Neutrophils

We examined the effect of the methanolic extract Vau-1, Vau-2 or pure arbutin (all dissolved in dimethyl sulfoxide (DMSO)) on survival of neutrophils isolated and purified from bone marrow (BM). We observed the decreased vitality of neutrophils cultured in the presence of 50 µg/mL Vau-1 or Vau-2 extracts in comparison to control culture (Figure 2B). We noticed that the vehicle DMSO (0.3%) itself can have a marginal stimulatory effect on neutrophil vitality (Figure 2B), hence, we further evaluated the activity of the extracts or arbutin in comparison to DMSO-treated cultures (Figure 2C,D). Vau-1 induced a dose-dependent decrease in neutrophil survival (Figure 2C). The treatment of neutrophils with Vau-1 led to a dose-dependent decrease in cell survival, with the lowest survival observed at the highest concentration of extract used, 50 µg/mL. Interestingly, pure arbutin at the same concentration also demonstrated an inhibitory effect, albeit less pronounced, on neutrophil survival, considering that 50 µg/mL Vau-1 extract contains 56.7 µM arbutin (Figure 2B; Table 2). At concentrations ranging from 4 to 15 µg/mL (14.7 to 55.1 µM), arbutin increased neutrophil survival; at higher concentrations from 61–245 µg/mL (224 to 900 µM), it decreased cell density and cell vitality (Figure 2D).

### 2.3. Effect of Vau-1/Vau-2 Extracts or Arbutin on TNF-α and IFN-γ Production in BM-Derived Neutrophils

Next, we delineated the production of the pro-inflammatory cytokines IFN-γ and TNF-α in BM-derived neutrophils, using flowcytometry and measuring mean fluorescence intensity (MFI) (Figure 3A,B). The presence of DMSO (0.3%) had a stimulatory effect on IFN-γ production that was reduced by both extracts Vau-1 and Vau-2 at concentrations < 50 µg/mL (corresponding to 56.7 µM arbutin in Vau-1 and 70.4 µM arbutin in Vau-2, respectively; see Table 2) (Figure 3A). Both Vau-1 and Vau-2 strongly inhibit the intracellular level of IFN-γ at a low concentration (0.025 µg/mL of the extracts contain 0.028 µM arbutin in Vau-1 and 0.035 µM arbutin in Vau-2, respectively; Table 2; Figure 3A). Arbutin had an opposite effect on intracellular IFN-γ by significantly inducing IFN-γ production when used at lower concentrations, ranging from 0.25 to 0.0025 µg/mL (corresponding to 0.9 to 0.009 µM, respectively) (Figure 3A). These data suggest that the action of arbutin on IFN-γ production might be masked by other molecules present in the Vau-1 and Vau-2 extracts, or that arbutin specifically interfered with phorbol 12-myristate 13-acetate and ionomycin (PMA/Yon) signaling.

When we investigated the intracellular level of TNF-α (Figure 3B), we found that Vau-1 stimulated, in a dose-dependent manner, this cytokine’s production. Vau-2 and arbutin inhibited the cytoplasmic TNF-α in comparison to DMSO group, suggesting that other molecules in the extract might contribute to this activity. To eliminate the possibility that Vau-2 and arbutin induce the release of the cytokine, rather than its accumulation in the cytoplasm, the neutrophils were stimulated with PMA/Yon in the presence of monensim.

### 2.4. Effect of Vau-1/Vau-2 or Arbutin on the Pattern of Immature or Mature Neutrophils Expressing TNF-α and IFN-γ

The bone marrow contains neutrophils at various stages of maturation, which can be distinguished in mice by the density of Ly6G expression and cellular granularity (Figure 4A). Ly6G^hi^ neutrophils (cells expressing the marker at high levels) were defined as mature, while Ly6G^low^ neutrophils (expressing the marker at low levels) were defined as an immature population (Figure 4A). Ly6G^+^ cells also showed variability in granularity content, when evaluated using the SSC-scatter on the flow cytometer. Immature cells were less granular and had a round shape, while mature neutrophils were identified as SSH^hi^ (Figure 4A), as they accumulated granules during differentiation. The total pool of BM-derived neutrophils contained sub-populations of cells with various shapes and maturity statuses (Figure 4A,B). In addition, each sub-population showed a particular pattern of cytoplasmic IFN-γ and TNF-α levels (Figure 5).

The vehicle DMSO (0.3%) increased the proportion of cells with an immature phenotype SSC^hi^Ly6G^low^ and SSC^low^Ly6G^low^ (Figure 4B). Vau-1/Vau-2 and arbutin all increased in a dose-dependent manner the immature transient pool, defined as SSC^hi^Ly6G^low^ (in blue), and decreased the mature transient pool, defined as SSC^low^Ly6G^hi^ (in green; Figure 4B). Vau-2 and arbutin, but not Vau-1 (at low concentration) expanded the SSC^low^Ly6G^low^ immature pool in comparison to the controls (in red; Figure 4B). The pool of SSC^hi^Ly6G^hi^ mature cells was neither markedly affected by the extracts, nor expanded significantly by arbutin at the concentrations used (purple; Figure 4B).

In our previous experiments (Figure 3), the extracts and arbutin changed IFN-γ and TNF-α cytoplasmic levels, but it is not clear if this effect is characteristic for a particular stage of neutrophil maturation. In the Ly6G^low^ state, neutrophils were single IFN-γ producers (Figure 5A), double TNF-α/IFN-γ producers or negative for both cytokines (representative dot plots at Figure 5A). The transition from Ly6G^low^ to Ly6G^hi^ was related to the loss of single IFN-γ producing cells and progressive expansion of the double TNF-α/IFN-γ positive neutrophils. Thus, we focused on the frequency of those double cytokine producers as an additional indicator for neutrophil maturation. We also distinguished the pattern of low and high IFN-γ expression that varied within the Ly6G^+^ sub-populations (Figure 5A). In control groups (control or DMSO treated), low intracellular IFN-γ level was characteristic for mature transient and mature states rather than immature pools (Figure 5A,B). The extracts and arbutin expanded in a similar manner the frequency TNF-α positive cells with lower IFN-γ expression (TNF-α^+^/IFN-γ^low^, Figure 5B), within the pool of mature transient state in comparison to control groups (in red in Figure 5B). Vau-1, at a low concentration (0.5 μg/mL), increased the proportion of TNF-α^+^/IFN-γ^low^ cells within the immature pool (in blue, Figure 5B), while Vau-2 (0.5 μg/mL) and arbutin (from 0.25 to 24.5 μg/mL (0.9–90 μM)) increased the proportion of those double producers in the immature transient pools (in green, Figure 5B). Vau-2 and arbutin, but not Vau-1, decreased, in a dose-dependent manner, the frequency of TNF-α^+^/IFN-γ^low^ producers within the mature pool (in purple, Figure 5B).

The frequency of double TNF-α^+^/IFN-γ^hi^ (Figure 5C) producers was higher in the extract- or arbutin-treated groups in comparison to controls. Vau-1 (0.5 μg/mL) further increased the proportion of those cells within the immature pool (in blue, Figure 5C), while Vau-2 and arbutin expanded, in a dose-dependent manner, the proportion of TNF-α^+^/IFN-γ^hi^ neutrophils with mature phenotype (purple, Figure 5C).

## 3. Discussion

In this study we performed chemical analysis of methanolic extracts from *V. austriaca* collected from two different habitats. The ^1^H NMR, in combination with 2D NMR spectroscopy, were used, and 13 individual compounds, including amino acids, organic acids, iridoid glucosides and phenolic compounds were identified. Among the secondary metabolites, the iridoid glucosides aucubin and catalpol, as well as the phenylethanoid glycoside—verbascoside found in both *V. austriaca* extracts are a characteristic feature for the *Veronica* plant species [2]. Previously, we observed that verbascoside acts as a potent modulator of neutrophil priming and activation, by altering various cellular functions, such as expression of the integrin CD11b and the chemokine CXCR2, and production of TNF-α and the degrading enzyme, MMP-9 [15]. At the molecular level, verbascoside can interfere with p38 signaling, while the compound aucubin can alter the metabolic AMP-activated protein kinase (AMPK) pathway, and can affect NRF2 activation [16]. The anti-inflammatory effect of catalpol was associated with an inhibition of neutrophil migration in a model of airway or paw inflammation [17]. Altogether, these studies suggested that the extracts of *V. austriaca* may exert potent biological activities, as they contain biologically active compounds, such as iridoid and phenolic glycosides.

We found that the major compound which comprised around 30% of the dry weight of the extracts was arbutin (in excess of 0.3 mg arbutin/mg extract). Although arbutin is widely distributed in the plant kingdom, its occurrence in *Veronica* spp. is not typical. To date, arbutin has been isolated and purified from *V. turrilliana* [18] and a commercial hybrid of *Veronica* [6], but has not been reported in *V. austriaca* [9]. The significant amount of arbutin in *V. austriaca* L. collected from two different habitats suggests that it was not produced coincidentally, due to different environmental conditions, for example. Arbutin-containing extracts of some plant species have, for centuries, been used in phytotherapy against urinary tract infection and skin hyperpigmentation, and have shown antioxidant, anti-inflammatory and antitumor activity [19]. Herein we observed for the first time that arbutin can affect neutrophil survival. The compound at concentrations ranging from 4 to 15 µg/mL (14.7 to 55.1 µM) increased neutrophil survival, while at higher concentrations from 61–245 µg/mL (224 to 900 µM), it decreased neutrophil density and vitality. Other studies in melanocytes have shown that arbutin is a potent tyrosinase inhibitor, with an IC_50_ value of 1.09 mM [20]. In A375 human malignant melanoma cells, arbutin at high concentration (1 mM) up-regulated 88 genes and down-regulated 236 genes, including genes (AKT1, CLECSF7, FGFR3, and LRP6) controlling cell cycle progression [21], hence, suggesting that it may act similarly on neutrophils. However, the effect on cell vitality of arbutin at low concentrations was unexpected because the compound was inactive at concentrations ranging from 5 to 20 μM in several screening libraries for anti-cancer drugs, agonists of the farnesoid-X-receptor (FXR) signaling pathway and cell viability, and antagonists of the sonic hedgehog signaling (Shh) pathway [2]. By contrast, the Vau-1 extract decreased neutrophil survival in a dose-dependent manner. Similar cytotoxic activity of *Veronica* species has been shown in various cell lines, such as Hep-2 (human epidermoid carcinoma), RD (human rhabdomyosarcoma), and L-20B (transgenic murine L-cells), where IC_50_ values were above 150 μM [22]. At low concentration, the methanolic extract of the aerial parts of *Veronica* spp. demonstrated cytotoxicity against colon (HF-6) and prostate (PC-3) human cancer cell lines [23]. Most of the authors linked the presence of iridoid and phenolic compounds in *Veronica* extracts (aucubin, catalpol, catalpol derivatives and verbascoside, among others) with its cytotoxic effect. Indeed, in our previous study, we found that verbascoside is an inhibitor of TLR2 and TLR4-mediated apoptosis of BM-derived neutrophils, although less potent than isoverbascoside. The enumeration of live cells in total BM cultures showed that, at high concentration (>160 µM), the compound reduced neutrophil numbers with 20–26%, and increased apoptosis up to 60% in death-sensitive conditions (Annexin V staining) [15]. Thus, we cannot exclude the possibility that the reduced vitality of neutrophils following treatment with the Vau-1 extract may be due to the presence of verbascoside.

Neutrophils produce pro-inflammatory cytokines like TNF-α and IFN-γ, that can be pre-stored in the cytoplasm and secreted upon stimulation (priming or activation). We observed that Vau-1 and Vau-2 inhibited strongly the intracellular level of IFN-γ at low concentrations, while arbutin had an opposite effect by inducing a significant increase in IFN-γ production. We suggest that the action of arbutin on the cytokines might be masked by other molecules present in the Vau-1 and Vau-2 extracts. It is also possible that the compounds interfered selectively with PMA/Yon-induced calcium signaling after stimulation for 4 h. The latter hypothesis is based on a study demonstrating that aqueous-acetone extracts of *Veronica* spp. (*V. teucrium*, *V. jacquinii*, and *V. urticifola)* have an inhibitory effect on calcium ionophore-stimulated platelets, resulting in the suppressed release of pro-inflammatory enzymes 12-lipoxygenase (12-LOX) and cyclooxygenase-1 (COX-1) and pro-inflammatory mediators, such as interleukin-8 (IL-8) and E-selectin [11].

Interestingly, while Vau-2 and arbutin inhibited TNF-α production, Vau-1 induced a dose-dependent increase in the cytokine’s intracellular level. The presence of verbascoside might contribute to this effect. Several studies have shown that verbascoside can alter, via TAK1, the degree of NF-kB activation, phosphorylation and nuclear translocation [24]. This compound inhibited LPS-stimulated TNF-α release by intraperitoneal macrophages [25], the level of TNF-α in bronchoalveolar lavage fluid (BALF) in LPS-induced lung inflammation [26], and in liver in LPS-induced immunological liver injury [27]. We previously found that verbascoside at low concentrations (<16 μM) increased LPS-induced TNF-α production and frequency of TNF-α^+^ BM-derived neutrophils via inhibition of p38 phosphorylation [15]. Similarly, here, we found that the Vau-1 extract, which contains less amounts of verbascoside than Vau-2, elevated the TNF-α intracellular level. It is also possible that aucubin, as well as the combination of different compounds in the extract, can act synergistically or antagonistically on TNF-α production. Another reason might be that TNF-α is already pre-stored in mature neutrophils and less cells synthesize TNF-α de novo. Thus, the effect of the compounds may require strong cell activation to inhibit cytokine production at the mRNA level [28]. With regard to the pure compound, arbutin, we found the inhibited generation of TNF-α in agreement with other reports using LPS-stimulated murine BV2 cells, where this effect was mediated by inhibited nuclear translocation and the transcriptional activity of NF-κB [29].

Neutrophils undergo differentiation and maturation in BM. The process of granulopoiesis is precisely regulated by mediators released in the periphery cytokines and chemokines, and by mediators secreted in the environment of the bone marrow niches. During differentiation, neutrophils acquire various types of secretory vesicles and granules, as well as store cytokines in the cytoplasm. Indeed, here we observe that the transition from Ly6G^low^ to Ly6G^hi^ neutrophils was associated with the loss of single IFN-γ producing cells and the progressive expansion of double TNF-α/IFN-γ positive neutrophils. At the terminal differentiation stage, neutrophils are mature and ready to express integrin and chemokine receptors in response to signals from the periphery, and in order to be mobilized into the blood. We found that Vau-1/Vau-2 and arbutin increased in a dose-dependent manner the immature transient pool defined as SSC^hi^Ly6G^low^, but they failed to dramatically alter the pool of mature cells. However, both Vau-2 and arbutin, but not Vau-1, expanded the immature pool. In addition, we observed that the *Veronica* extracts and arbutin changed IFN-γ and TNF-α cytoplasmic levels in cells at a particular stage of neutrophil maturation. Vau-1 at low concentration increased the proportion of TNF-α+IFN-γ^low^ cells within the immature pool, while Vau-2 and arbutin elevated the proportion of those double producers in the immature transient pools, likely leading to the expansion of the proportion of TNF-α+IFN-γ^hi^ neutrophils with a mature phenotype. It has been shown that, during differentiation and maturation, cells require the presence of reactive oxygen species (ROS), a particular level of nitric oxide (NO) and functional antioxidant enzymes and mechanisms. Indeed, ROS are considered as intracellular messengers that interact with specific receptors, signalling pathways including protein kinases, phosphatases, and transcription factors [30]. Recent studies showed that methanolic, ethanolic and aqueous extracts from a variety of *Veronica* species possess powerful radical scavenging activity against superoxide (SO), and nitric oxide (NO) radicals [31,32]. Hence, we further speculate that Vau-2 and arbutin affected maturation and frequency of TNF-α+IFN-γ^hi^ mature neutrophils, by influencing the cellular antioxidant/oxidant mechanism, and perhaps by sustaining the activation of particular pathways and transcription factors. Arbutin may also be able to affect cell differentiation because it suppresses the tyrosinase elevation at the late stage of melanocyte differentiation [33]. It inhibits tyrosinase, an enzyme which can alter protein function by oxidation of the tyrosine residues, and starts further non-enzymatic reactions with other tyrosines, cysteines, lysines or histidines, at the same or a different molecule, resulting in inter- and/or intramolecular cross-links [34]. However, further investigations are needed to confirm the effect of arbutin and *Veronica* extracts on neutrophil differentiation and, therefore, to justify their application in acute inflammatory conditions or severe immunosuppression—both conditions with abrogated granulopoiesis and neutrophil maturation. 

## 4. Materials and Methods

### 4.1. Plant Material

Aerial parts from *V. austriaca* were collected in their flowering period in July 2019 in the Central Balkan Mts. (Vau-1; 42°47′19.90″ N 24°33′52.24″ E, 1592 m a.s.l.) and Rhodopi Mts. (Vau-2; 42°7′26.93″ N 24°24′49.52″ E, 429 m a.s.l.), and identified by Dr. Ina Aneva (Institute of Biodiversity and Ecosystem Research, Bulgarian Academy of Sciences). Voucher specimens SOM 1392 and SOM 1393, respectively, are hence deposited in the Herbarium of the Institute of Biodiversity and Ecosystem Research.

### 4.2. Extraction Protocol and NMR Analysis

Ground aerial parts of *V. austriaca* were air dried and 50 mg of each of 4 biological replicates were homogenized with equal amounts of CD_3_OD (0.75 mL) and D_2_O (0.75 mL KH_2_PO_4_ buffer, pH 6.0), containing 0.005% (*w*/*v*) trimethylsilyl propanoic acid (TSPA-*d*_4_). After 20 min ultrasonication (35 kHz; UCI-50Raypa^®^ R. Espinar S.L., Barcelona, Spain), samples were centrifuged (14,000× *g*, 20 min), then, the supernatants were transferred to thin glass walled tubes (NMR tube; 5 mm) and finally analyzed at the NMR spectrometer as described in [35]. Briefly, proton (^1^H) as well as 2D NMR spectra (*J-*resolved, COSY, HSQC), were recorded at 25 °C on an AVII+ 600 spectrometer (Bruker, Karlsruhe, Germany), operating at a proton NMR frequency of 600.01 MHz [35]. Deuterated methanol was used for internal lock. The resulting ^1^H NMR spectra for each sample was further processed by referencing to the internal standard TSPA, phased and baseline corrected, by running TopSpin software (3.6.5, Bruker BioSpin Group). CD_3_OD and D_2_O from Deutero GmbH (Kastellaun, Germany) were used in the experiments.

### 4.3. Extraction Protocol for HPLC Analysis and Biological Tests

Air dried ground plant material (5 g) of each sample was homogenized with 125 mL of absolute CH_3_OH. After 60 min ultrasonication, the extract was filtered and evaporated to dryness in vacuo. The yields of the extract were 0.78 g (Vau-1) and 0.83 g (Vau-2).

### 4.4. HPLC Analysis of Arbutin Content in V. austriaca Extracts

The HPLC analyses were performed according to Rathi et al., 2019 [36], with some modifications, as described below. The quantification was conducted on Waters HPLC system, consisting of Waters 1525 Binary pumps with Waters 2487 Dual λ Absorbance Detector (Waters, Milford, MA, USA), equipped with a reverse-phase Kinetex^®^ C_18_, 100 Å (150 × 4.6 mm, 5 µm) core-shell column (Phenomenex, Torrance, CA, USA), operating at 26 °C. The mobile phase consisted of water (solvent A) and acetonitrile (solvent B), with a flow rate 0.5 mL/min and gradient elution as follows: 0–2 min 99% A; 2–6 min decreased to 40% A; 6–15 min gradually increased back to 99% A arbutin was detected at 285 nm. Six solutions of the arbutin, with different concentrations ranging from 25 to 200 μg/mL dissolved in methanol, were used to construct a linear calibration curve.

### 4.5. Animal Studies

Balb/c mice were purchased from Charles River Laboratories (Wilmington, MA, USA) and then bred in the Experimental Animals Facility at the Institute of Microbiology (Sofia, Bulgaria). Mice (female, 8-week-old, 18–20 g weight) were kept under standard conditions, fed with a laboratory diet (29% protein, 13% fat, 56% carbohydrates) and given water ad libitum, as described previously [13]. The experiments were done under anesthesia and the control of a veterinarian. They were approved by National Food Agency (Sofia, Bulgaria), according to National and European Guidelines (License for Animal Housing No 352/30.01.2012 (registration No11130005); License for Experimental Procedures No 105/10.07.2014). All animal experiments were conducted according to ARRIVE (Animal Research: Reporting of In Vivo Experiments).

### 4.6. Isolation of Bone-Marrow-Derived Neutrophils

Bone marrow (BM) cells were isolated from the femur and tibia (both legs) of healthy Balb/c mice (n = 7, female, 8-month-old, weight 18–22 g), as previously described [14]. The BM cell suspension was subjected to 63% Percoll gradient in 0.15 M NaCl (Percoll refractory index 1.3450; density 1.090 g/mL (GE Healthcare, Sigma-Aldrich, Darmstadt, Germany)), in centrifugation at 500× *g*, 40 min, room temperature. This was followed by erythrocyte lysis of the sediment with an ammonium-chloride-potassium (ACK) buffer (150 mM NH_4_Cl, 10 mM KHCO_3_, 0.1 mM Na_2_EDTA, all chemicals from Sigma-Aldrich) Finally, the cells were washed with phosphate-buffered saline (PBS; pH 7.4) at 250× *g* for 10 min, counted and resuspended at 2 × 10^6^/mL in sterile complete Roswell Park Memorial Institute Medium (RPMI)-1640 medium, containing 10% Fetal Calf Serum (FCS), 2 mM L-glutamine, 100 U/mL penicillin, 100 µg/mL streptomycin (all from Sigma-Aldrich, Darmstadt, Germany). The isolated cell population consists of > 90% viable cells, of which 88–90% are positive for Ly6G and 90% are mature cells with banded/segmented nuclei, as assessed by Giemsa staining.

### 4.7. Cell Vitality

BM-derived neutrophils were resuspended at the concentration 2.4 × 10^6^/mL in 10% FCS/RPMI medium, and seeded in a volume of 100 µl on a sterile 96 well-plate (Corning, Wiesbaden, Germany). Cells were cultured in the presence of the methanolic extract Vau-1 or Vau-2 or arbutin at concentration 50 µg/mL, all dissolved in DMSO. The pure compound arbutin was isolated previously from the methanolic extract of *V. turrilliana* [18] (Figure 2A, chemical structure). In some experiments, the cells were cultured in the presence of increasing concentrations of Vau-1/Vau-2, ranging from (0.025 to 1000 μg/mL) or of arbutin at concentrations ranging from 0.025 to 490 μg/mL (0.09–1800 μM). After 36 h the cells were fixed, stained with Janus Green dye (Abcam, Cambridge, UK) and washed 3 times with dH_2_O. The dye was eluted by washing with elution buffer (Abcam, Cambridge, UK). The OD 630 nm was measured in samples in triplicate. Janus Green is a basic vital stain that determines cell density and specifically stains the mitochondria in living cells. The vitality was calculated as a percentage of control and according to the formula below:Survival = 1−((OD_630_ DMSO – OD_630_ Control)/(OD_630_ drug−OD_630_ Control))

### 4.8. Intracellular Flowcytometry

BM-derived neutrophils or BM cells were resuspended at a concentration of 1 × 10^6^/mL in 10% FCS/RPMI, and seeded in a volume of 500 µL on 24 well plates (Corning, Wiesbaden, Germany). Cells were cultured in the presence of increasing concentrations of the Vau-1 or Vau-2 extracts or arbutin for 36 h. At the last 4 h of culture, cells were stimulated with phorbol-12-myristate 13-acetate (PMA; 100 ng/mL) and ionomycin (Yon 10 µg/mL; PMA/Yon, both from Sigma-Aldrich, Darmstadt, Germany), in the presence of the Golgi inhibitor monensim (1 µM in DMSO; Sigma-Aldrich, Darmstadt, Germany), to prevent cytokine degranulation. Cells were washed with 2% bovine serum albumin (BSA, Sigma-Aldrich, Darmstadt, Germany) in PBS containing 2 mM ethylenediaminetetraacetic acid (EDTA), stained for the expression of the murine neutrophil marker Ly6G, using anti-Ly6G antibody labeled with Fluorescein isothiocyanate (FITC) (clone A1; Biolegend, London, UK), washed with PBS, and fixed with 4% paraformaldehyde (PFA)/PBS (Thermo Fisher Scientific, Stuttgart, Germany) for 10 min at room temperature. The cells were then incubated with permeabilization buffer (Biolegend, London, UK) for 10 min, followed by incubation with antibodies diluted in 5% BSA/PBS/EDTA at concentration 0.1 μg/mL per 1 × 10^6^ cells, against IFN-γ (labeled with PE/Cy7) and anti-TNF-α (labelled with APC/Cy7; both from Biolegend, London, UK) and isotype controls (Biolegend, London, UK) for 1 h at 4 °C. After washing with PBS, the cells were subjected to flowcytometry analysis for intracellular cytokine production. The mean of fluorescence intensity (MFI) and the number of positive cells (single or double) in the live Ly6G^+^ population were used in the data analysis, counting at least 30,000 events (BD FCSDiva v6.1.2 Software (Becton Dickinson GmbH, San Jose, CA, USA) and Flowing Software 2.2 (Cell Imaging Core, Turku Centre for Biotechnology, Turku, Finland). In order to determine whether or not cytokine production depends on the stage of differentiation and maturity of BM neutrophils, a more detailed analysis was performed on gated SSC^hi^ and SSC^low^ and Ly6G^hi^ and Ly6G^low^ populations, as follows (See Figure 4A):

Upper Left—gate P1: SSC^hi^Ly6G^low^—neutrophils at immature transient state

Upper right—gate P2: SSC^hi^Ly6G^hi^—neutrophils at mature state

Lower left—gate P3: SSC^low^Ly6G^low^—neutrophils at immature state

Lower right—gate P4: SSC^low^Ly6G^hi^—neutrophils at mature transient state

### 4.9. Statistical Analyses

A statistical analysis was accomplished by using InStat 3.0 (GraphPad Software, La Jolla, CA, USA). Data were presented as mean ± standard deviation (SD). The differences in the mean values between groups were analyzed with the two-tailed Student *t*-test. Differences were considered significant when *p* < 0.05.

## 5. Conclusions

In the present study, we observed, for the first time, that the methanolic extracts of *V. austriaca* collected from two different habitats contained arbutin as a major constituent of the secondary metabolites pool. The Vau-1 extract inhibited, in a dose-dependent manner, neutrophil vitality, increased the intracellular level of TNF-α, and the proportion of TNF-α+IFN-γ^low^ cells within the immature pool, while Vau-2 extract inhibited IFN-γ and TNF-α production, but expanded the frequency of mature TNF-α+IFN-γ^hi^ within the BM pool. Arbutin, at low concentrations, increased neutrophil survival, elevated the proportion of double TNF-α/IFN-γ producers in the immature transient pool, and increased the frequency of mature TNF-α+IFN-γ^hi^ cells in the BM population. As *Veronica* plant species are widely used in traditional medicine, the identification of the plant metabolites contributing to their biological activity is of significant importance, and can eventually lead to the formulation of pharmaceutically relevant molecules.

## Figures and Tables

**Figure 1 molecules-25-03410-f001:**
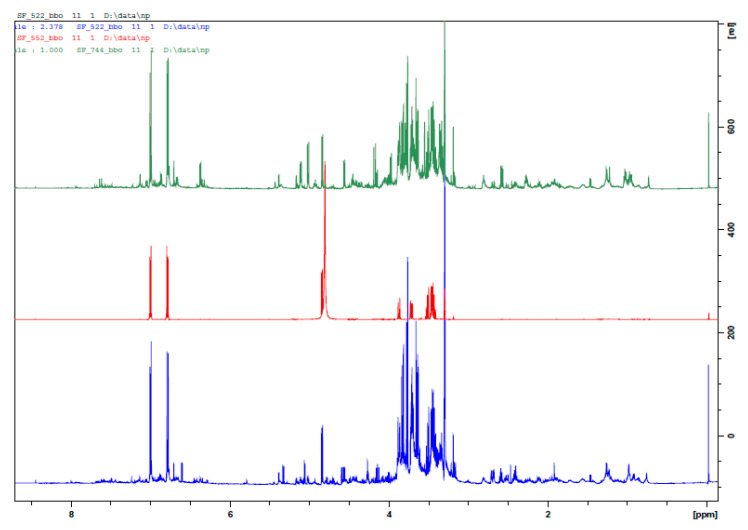
^1^H NMR spectra of Vau-1 (blue), pure arbutin (red) and Vau-2 (green).

**Figure 2 molecules-25-03410-f002:**
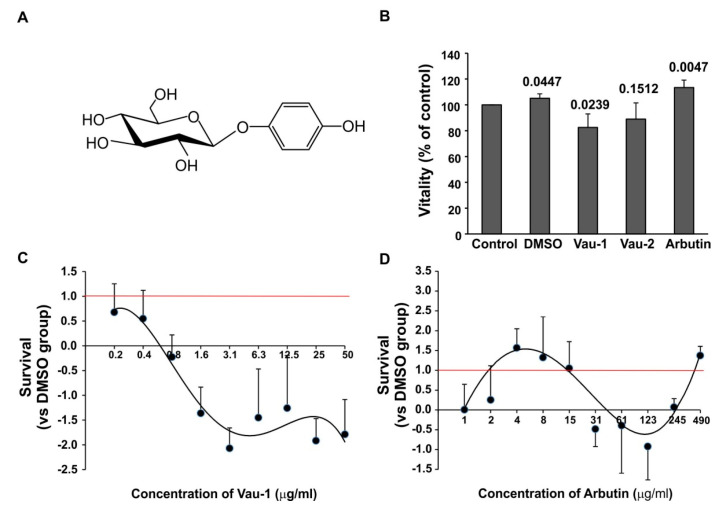
Effect of the Vau-1/Vau-2 extracts or arbutin on neutrophil vitality and survival. (**A**) Chemical structure of arbutin; (**B**) Vitality of bone marrow (BM)-derived neutrophils compared to control cultures. Vitality was calculated as a percentage of control cultures containing cells only. Data represent mean ± SD of cell samples isolated from 7 mice and plated in triplicate. P values are shown for each group when compared to control; (**C**) Dose-dependent effect of Vau-1 on DMSO-induced survival of BM-derived neutrophils. Data represent mean ± SD of sample isolated and pooled from 7 mice and run in triplicate. Red line shows the survival (relative value) at DMSO-treated group. Black line is the polygonal trend line drawn to extrapolate the vitality vs. Vau-1 extract concentration; (**D**) Dose-dependent effect of arbutin on DMSO-induced survival of BM-derived neutrophils. Data represent mean ± SD of samples isolated from 7 mice and plated in triplicate. Red line shows the survival (relative value) at DMSO-treated group. Black line is the polygonal trend line drawn to extrapolate the vitality vs. the arbutin concentrations.

**Figure 3 molecules-25-03410-f003:**
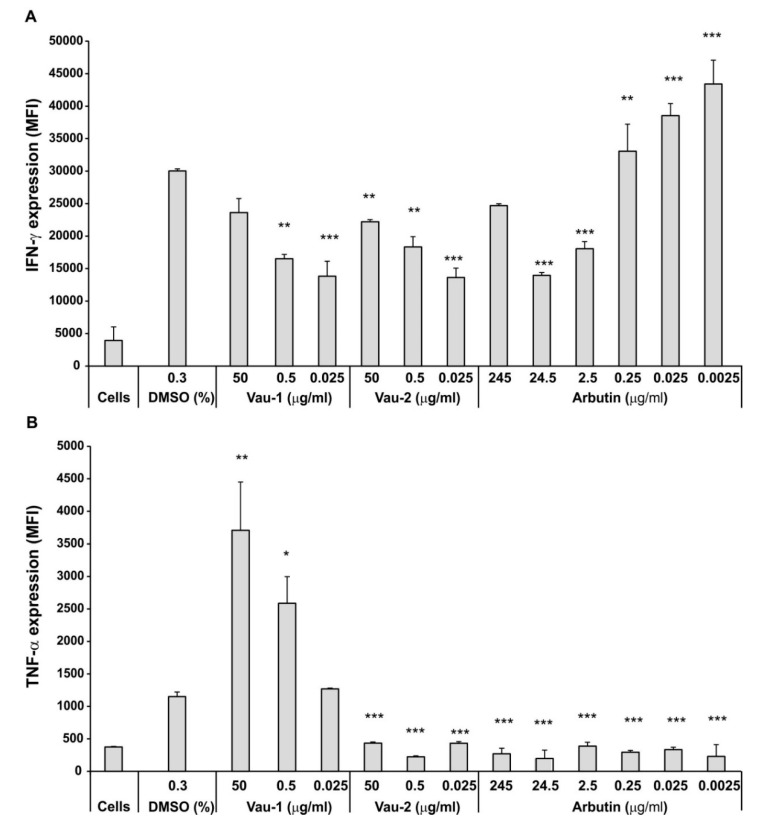
Effect of Vau-1/Vau-2 or arbutin on production of cytokines IFN-γ (**A**) and TNF-α (**B**). Purified neutrophils were cultured in the presence of DMSO (0.3%) and decreasing concentrations of Vau-1, Vau-2 and arbutin for 36 h. Control cells were incubated with phosphate-buffered saline (cells). In the last 4 h of incubation, the neutrophils were stimulated with PMA/Yon in the presence of the Golgi inhibitor, monensim, in order to maximize cytokine accumulation. The neutrophils were then washed and stained for the neutrophil marker, Ly6G. The cells were then fixed, permeabilized and incubated with PE/Cy7 or APC/Cy7 conjugated antibodies against TNF-α and IFN-γ. After washing, gated Ly6G^+^ cells were subjected to flow cytometry analysis for the intracellular production of cytokines. Data represent mean ± SD of sample from 2 experiments with neutrophils isolated and pooled from 7 mice and assayed in duplicate. P-values are shown for each group when compared to the group cultured with 0.3% DMSO; * *p* < 0.05; ** *p* < 0.01; *** *p* < 0.001, two-tailed Student *t*-test.

**Figure 4 molecules-25-03410-f004:**
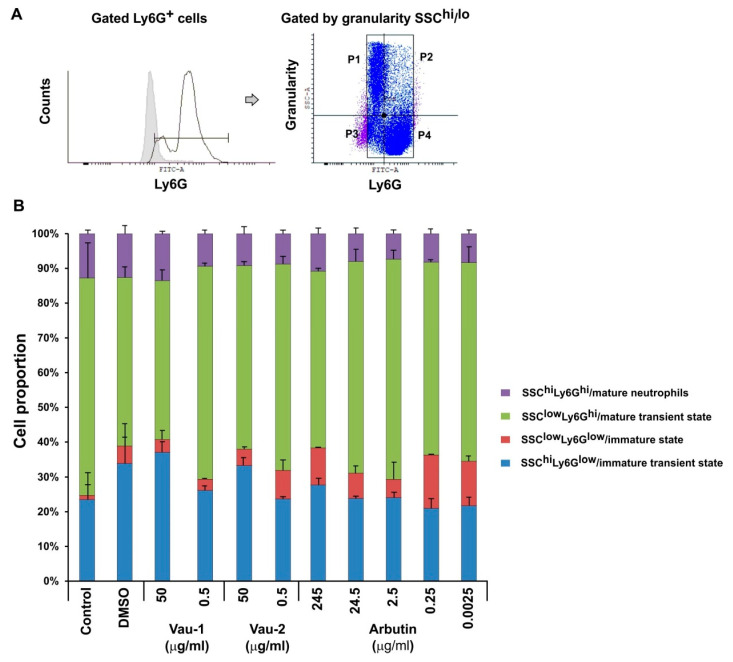
Effect of Vau-1/Vau-2 extracts or arbutin on proportion of neutrophils at various maturation states. (**A**) Gating strategy for neutrophils. The first panel shows the histogram for Ly6G expression and gating of Ly6G^+^ positive cells. The second panel shows the dot-plot histogram with scales for Ly6G^+^ cells vs. SSC-A (the size surface volume), that determines the shape and granularity of the BM-derived neutrophils. The following populations were defined on the basis of SSC-A and Ly6G positivity (staining with fluorescein isothiocyanate (FITC)-labelled antibody against Ly6G). Upper Left (P1 gate)—SSC^hi^Ly6G^low^ immature transient cells; Upper right (P2 gate)—SSC^hi^Ly6G^hi^ mature cells; Lower left (P3 gate)—SSC^low^Ly6G^low^ immature cells; Lower right (P4 gate)—SSC^low^Ly6G^hi^ mature transient cells; (**B**) Proportion of neutrophils at various maturation states in BM cells incubated in the presence or absence of Vau-1, Vau-2 or arbutin for 36 h. Data represent mean ± SD of sample from 2 experiments with neutrophils isolated and pooled from 7 mice and assayed in duplicate.

**Figure 5 molecules-25-03410-f005:**
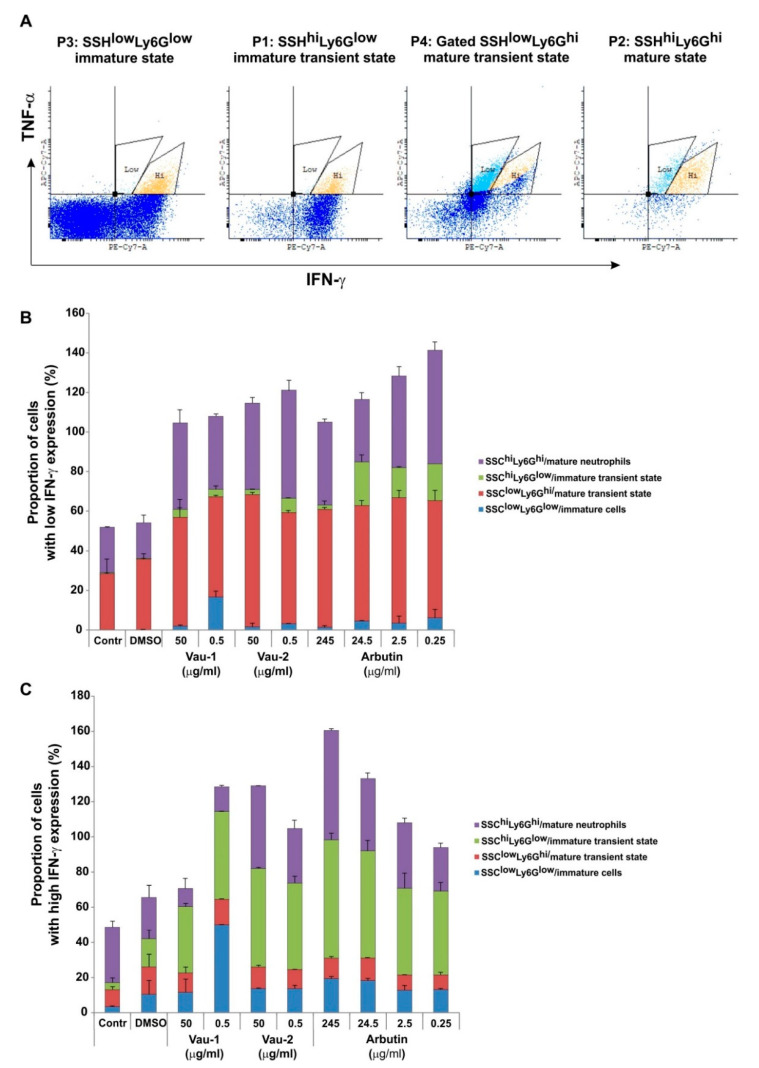
Effect of Vau-1/Vau-2 extracts or arbutin on the frequency of double TNF-α^+^/IFN-γ^+^ producers within the pool of immature and mature neutrophils in BM. (**A**) Representative dot-plot histograms showing two populations of double TNF-α^+^/IFN-γ^+^ producers in the control group: TNF-α^+^/IFN-γ^low^ and TNF-α^+^/IFN-γ^hi^ which varied in frequency in immature, mature or transient immature or mature pools; (**B**) Proportion of TNF-α^+^/IFN-γ^low^ neutrophils (in %) at various maturation state in BM cell cultures incubated in the presence or absence of Vau-1, Vau-2 or arbutin for 36 h. Data represent mean ± SD of sample from 2 experiments with neutrophils isolated and pooled from 7 mice and assayed in triplicate; (**C**) Proportion of TNF-α^+^/IFN-γ^hi^ neutrophils (in %) at various maturation state in BM cell cultures incubated in the presence or absence of Vau-1, Vau-2 or arbutin for 36 h. Data represent mean ± SD of sample from 2 experiments, with neutrophils isolated and pooled from 7 mice and assayed in triplicate.

**Table 1 molecules-25-03410-t001:** *V. austriaca* L. metabolites, identified by relevant 1D and 2D NMR spectra.

Metabolite	Vau-1 ^1^	Vau-2 ^1^	Selected Signals, Multiplicity and Coupling Constant ^2^
Alanine	+	+	δ 3.71 (q)/δ 1.48 (d, *J* = 7.4)
α-Glucose	+	+	δ 5.18 (d, *J* = 3.8)/δ 3.49 m
β-Glucose	+	+	4.57 (d, *J* = 7.9)/3.19 (dd, *J* = 7.9, 9.2 )
Sucrose	+	+	δ 5.40 (d, *J* = 3.9)
Acetic acid	+	−	δ 1.93 (s)
Lactic acid	+	+	δ 4.06 m/δ 1.27 (d, *J* = 6.5)
Succinic acid	+	+	δ 2.47 (s)
Formic acid	+	+	δ 8.46 (s)
Choline	+	+	δ 3.21 (s)
Arbutin	++++	++++	δ 7.00 (d, *J* = 8.9)/δ 6.79 (d, *J* = 8.9)/δ 4.85 (d, *J* = 7.7)/δ 3.87 (dd, *J* = 12.3, 2.0)/δ 3.72 (dd, *J* = 12.3, 5.4)
Aucubin	++	+	δ 6.29 (dd, *J* = 6.2, 1.9)/δ 5.79 (t, *J* = 1.7)/δ 5.09 (dd, *J* = 6.3, 2.4)/δ 5.07 (d, *J* = 6.1)/δ 4.70 (d, *J* = 7.9)
Catalpol	+	++	δ 6.37 (dd, *J* = 6.0, 1.8)/δ 5.11 (dd, *J* = 6.0, 4.6)/δ 5.03 (d, *J* = 9.8)/δ 4.18 (d, *J* = 13.3)/δ 3.98 (dd, 8.3, 1.3)/δ 3.78 (d, 13.3)/δ 3.55 (brs)/δ 2.58 (dd, *J* = 9.8, 7.7)/δ 2.27 m
Verbascoside	+	++	δ 7.63 (d, *J* = 15.9)/δ 7.14 (d, *J* = 2.0)/7.05 (dd, *J* = 8.3, 2.0)/δ 6.67 (dd, *J* = 8.3, 2.0)/δ 6.34 (d, *J* = 15.9)/4.93 (t, *J* = 9.6)/4.47 (d, *J* = 7.9)/δ 2.81 (t, *J* = 7.2) 1.04 (d, *J* = 6.4)

^1^ The number of ‘‘+’’ refers to relative fold differences and ‘‘−’’ to absence of the particular compound. ^2^ Proton NMR chemical shifts (δ) and coupling constant (*J*).

**Table 2 molecules-25-03410-t002:** Calculations for arbutin concentrations in *V. austriaca* L. extracts.

Concentration of Extracts	Arbutin in Vau-1		Arbutin in Vau-2	
µg/mL	µg	µM	µg	µM
1000.0	308.2	1133.1	382.9	1407.7
795.0	245.0	900.8	304.4	1119.1
500.0	154.1	566.5	191.5	703.9
250.0	77.1	283.3	95.7	351.9
125.0	38.5	141.6	47.9	176.0
62.5	19.3	70.8	23.9	88.0
50.0	15.4	56.7	19.1	70.4
31.3	9.6	35.5	12.0	44.1
25.0	7.7	28.3	9.6	35.2
15.6	4.8	17.7	6.0	22.0
12.5	3.9	14.2	4.8	17.6
7.81	2.4	8.8	3.0	11.0
3.91	1.2	4.4	1.5	5.5
3.13	1.0	3.5	1.2	4.4
1.95	0.6	2.2	0.7	2.7
0.98	0.3	1.1	0.4	1.4
0.78	0.2	0.9	0.3	1.1
0.50	0.2	0.6	0.2	0.7
0.25	0.1	0.28	0.1	0.35
0.025	0.01	0.028	0.01	0.035

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
