# Peer review of "Veronica austriaca L. Extract and Arbutin Expand Mature Double TNF-α/IFN-γ Neutrophils in Murine Bone Marrow Pool"

_molecules, 2020, doi:10.3390/molecules25153410_

Round 1

Reviewer 1 Report

The article describes the identification of arbutin in extracts from the Veronica austriaca plant, which is a novel finding. Various techniques were used to investigate the extracts of two plant habitats and pure arbutin for comparison. This paper is very interesting and relevant for publication in Metabolites. However, there are some inconsistencies in units used for Vau-1, Vau-2, and arbutin that should be standardized for clarity and easier comparison of results. Specific comments and questions for the text are listed below.

Line 53: Define acronym “LC-MS/MS” at first appearance in text.

Line 69, Table 1: Explain in legend what multiple plus signs mean. Do these indicate relative abundance? If so what are the criteria? Why do you think acetic acid is + in Vau-1 but – in Vau-2?

Lines 86-92: This seems like information for the materials and methods section, not results.

Line 102, Figure 1: Why are only results for Vau-1 shown and not Vau-2? Were results similar for Vau-2? I suggest plotting the Vau-2 results on same graph in 2C in a different color for comparison. I would expect these to be similar, so it would be good to show confirmation of the trend. If there are no results for Vau-2 please explain why.

Line 102, Figure 1, 2C and 2D: why are the units for 2C and 2D different? (ug/mL for 2C vs. uM for 2D)? This makes visual comparison of the results difficult. Please change one of the figures to have the same units as the other for easier comparison of the data. It is difficult to tell if trends for Vau-1 and arbutin are similar or different. If they are different, what is the hypothesis for this?

Lines 104-120: This figure legend is very long, and much of this information can be included in the Materials and Methods section and referenced here. Information regarding the figures themselves (explanation of red lines vs. black lines and statistics) should be kept in legend, but information on how neutrophils were cultured is unnecessary here.

Lines 111-112 and 117: Again, use consistent units for 2C and 2D- I cannot tell without calculation if the range of concentrations for 2C (0.19-50 ug/mL) is similar to 2D (3.5-1800 uM); and if it is not the same, then why?

Lines 124-190: Seems some of this information should be in Materials and Methods, and the section can be references in Results.

Line 146, Figure 3: Why are results for Vau-1 and Vau-2 shown in ug/mL while arbutin is shown in uM? This makes it very difficult to compare results. Please change figure so that units are all the same for Vau-1, Vau-2, and arbutin.

Lines 155-156: Were corrections applied to statistical analyses for performing multiple tests?

Line 169, Figure 4: Again, not sure why Vau-1 and Vau-2 are in different units than arbutin. Please correct.

Line 175: Define “FITC” at first use in text.

Line 180, Figure 5: Correct units.

Line 192: Superscript “lo” should be “low”

Lines 197 and 214: “Lila”- should this be “lilac” or just “purple”?

Lines 269-271: Yes, the Vau-1 and Vau-2 appear to show differences compared to arbutin in the figure, but it is difficult to discern the relevance since concentrations of Vau-1 and Vau-2 are not the same as for arbutin. Comparisons of Vau-1 and Vau-2 with arbutin should be performed at the same concentrations for direct comparisons. Otherwise, how do we know that there are true differences or just differences in concentration that make it appear more significant than it really is?

Line 290: “raison” – do you mean “reason”?

Line 355: “preformed” – do you mean “performed”?

Line 361: “The flow rate decreased to 40% A”- I think you mean the mobile phase composition changed, or did you also change the flow rate from 0.5 mL/min? Please clarify.

Line 363: “conduct a calibration curve”- I think you mean “construct” a calibration curve?

Lines 362-363: Did you force calibration curve through 0 for calibration or was there an off set? What type of curve fit was used, linear?

Line 367, animal studies: Only female mice are mentioned, but in line 375 it states that male mice were also used. If male mice were used, also mention in section 4.5. How many of the 7 mice were male versus female?

Lines 376-377: missing and end parenthesis from (0.15 M NaCl.

Line 378: Define “RT”. I believe that you mean “room temperature” but RT is also used for “retention time” in scientific articles, so I would define at first use.

Line 379: Define “ACK” at first use.

Line 381: Define “RPMI” at first use.

Line 389: Do you mean that cells were cultured with Vau-1, Vau-2, and arbutin all together (text states AND), or do you mean cells were cultured separately with either Vau-1, Vau-2, OR arbutin? I think that it should be OR according to the text.

Line 422: Be consistent in whether you use superscript of “lo” or “low” throughout text.

Reviewer 2 Report

The aim of the manuscript ID 882095 was to evaluate the polyphenol profile of the Veronica austriaca aerial parts and their effect on the survival of neutrophils isolated from murine bone marrow. After reviewing, my comments concerning the manuscript are general positive, but some areas that need to be addressed before being suitable for publication is:

  • In the Introduction, the Authors provided the information on the chemical composition of V. austriaca but in my opinion this issue is too briefly presented. I suggested, that more information and references about types of isolated polyphenols of subjected V. austriaca should be included in the Introduction. The Authors proved in the present study that arbutin is the dominant component of the examined extracts. On the basis of the literature data, it can also be concluded that phenolic glycosides constitute the dominant fraction of polyphenols identified in the aerial parts of V. austriac?
  • Arbutin, identified in the analyzed extracts by 1H NMR method, was also quantified by HPLC. Please provide the HPLC chromatogram of the polyphenol separation of the studied extracts. What is the content of the other phenolic compounds such as the iridoid glycosides, phenylethanoid glycosides. Among the identified polyphenols, were there also acylated flavonoids?
  • Regarding the evaluation of the extracts and arbutin effects on the survival of neutrophils isolated from the mouse bone marrow and on the production of cytokines - tumor necrosis factor TNF-α and interferon IFN- g, please evaluate the effectiveness of the tested extracts by comparing obtained results with positive standards.
